# Numerical Simulation of a New Flow Field Design with Rib Grooves for a Proton Exchange Membrane Fuel Cell with a Serpentine Flow Field

**Xin Luo [1], Shizhong Chen [1,\*], Zhongxian Xia [1,\*] [ID], Xuyang Zhang [2] [ID], Wei Yuan [1] and Yuhou Wu [1]**

1   School of Mechanical Engineering, Shenyang Jianzhu University, 25 Hunnan Middle Road, Hunnan District, Shenyang, Liaoning 110168, China; mrluo0317@163.com (X.L.); 13751195427@163.com (W.Y.); wuyh@sjzu.edu.cn (Y.W.)
2   Clean Energy Research Institute, College of Engineering, University of Miami, Coral Gables, Miami, FL 33146, USA; x.zhang18@umiami.edu
\*   Correspondence: chensz@sjzu.edu.cn (S.C.); xiazx17@sjzu.edu.cn (Z.X.)

**Abstract:** The cathode flow field design of a proton exchange membrane (PEM) fuel cell is essential to fuel cell performance, which directly affects the uniformity of reactant distribution and the ability to remove water. In this paper, the single serpentine flow field design on the cathode side is optimized to reach a high performance by controlling the rib groove rate (the ratio of the number of grooved ribs to the number of total ribs). The rib groove starts from the inlet side and then evenly distributes over the ribs. Four rib groove rates are selected in this study, namely, 0, 1/3, 2/3, and 1. A three-dimensional PEM fuel cell model is used to analyze the output performance of the fuel cell. The results indicate that the rib groove design has a significant effect on the distribution of oxygen at the cathode side, the density of the membrane current, the concentration of water vapor under the rib, and the fuel cell output performance. The output performance of the fuel cell improves with the increased rib groove rate. However, when the rib groove rate is greater than 2/3, its impact on the overall performance of the fuel cell begins to slow down. The PEM fuel cells exhibit the best output performance when the rib groove rate is 1.

**Keywords:** proton exchange membrane fuel cell; cathode flow field design; rib groove; COMSOL Multiphysics; current and power density curves

## 1. Introduction

A proton exchange membrane (PEM) fuel cell is a power generation device which directly converts the chemical energy stored in fuel and oxidants into electric energy via electrochemical reactions. Its advantages include high energy conversion efficiency, no emissions of nitrogen oxides and sulfur compounds, low operating noise, and high reliability. As one of the important components of a PEM fuel cell, flow field accounts for about 60–80% of the total weight of the fuel cell and 30–50% of the total cost [1–3], which are key factors that can influence the output performance and cost of the fuel cell. Thus, this area is one of the focuses of current scientific research. At the macro level, the structure of the flow field has two main functions. On the one hand, it provides fuel and oxidants for the electrochemical reaction of the whole cell and discharges unreacted gas and reaction products (mainly water); on the other hand, the flow field collects the current generated by the electrochemical reaction. In brief, the structure of flow field has a great effect on the output performance of PEM fuel cells, and thus affects the commercialization. Although many types of flow fields have been studied and reported in the literature, such as parallel flow field [4–8], serpentine flow field [4,9,10], interdigitated flow field [11,12], and so on, every type has advantages and disadvantages. However, there are still

many challenges to be overcome to enhance the performance of flow fields, therefore, it is necessary for researchers to study and optimize the flow field structure.

In an experimental simulation of PEM fuel cell flow field, Ashorynejad et al. [13] designed novel flow field "speed bumps" to generate pressure fluctuation in the flow passage, and then numerically simulated its impact on PEM fuel cell performance. The results showed that "speed bumps" improved the polarization curve performance of proton exchange membrane fuel cells (PEMFCs). Alrwashdeh et al. [14] and Kim et al. [15] proved with their experiments and simulations that the protruding part of the flow field significantly improved the drainage capacity and the efficiency of oxygen transfer to the catalyst layer, so as to improve the performance of PEM fuel cells at medium and low potentials. Zhang et al. [16] simulated the flow path and protrusion of the flow field and found that the oxygen transfer capacity and drainage capacity were significantly improved. Heidary et al. [17] studied the different performances of PEM fuel cells in parallel flow fields under different blocking conditions. Their simulation results demonstrated that the performance of parallel flow fields with alternating blockage was not different from that of a serpentine flow field. Manso et al. [7] and Liu et al. [18] simulated the influence of changing flow field lengths and cross-sections on PEM fuel cell performance. The results showed that these two factors played important roles in the design of PEM fuel cell flow fields.

Wang et al. [19,20] proposed a new shunt design for a parallel cathode flow field, and studied the polarization curve, pressure drop, cathode oxygen distribution, and water management of the fuel cell via experiments and simulations, and then optimized and analyzed the shunt design. Jiao et al. [21] studied the water transfer and discharge in a flow field by using a hydrophilic field plate. The results indicated that the hydrophilic field plate was useful in the discharge of water, which is of great significance for PEM fuel cell water management. By means of numerical simulation, Caglayan et al. [22] studied the effect of temperature on a high-temperature proton exchange membrane (HT-PEM) fuel cell output power, with results showing that the output power of the HT-PEM fuel cell increased with increasing temperature. At the same time, the pressure drop variation in the anode and cathode channels, the oxygen concentration distribution in the cathode channel, and the water concentration distribution in the cathode channel were predicted at 160 °C. Bachman et al. [8] studied the lateral diffusion of substances in the back of a flow field by adding valves at some outlets of the parallel flow field. In addition, the experimental platform was improved to study water migration characteristics and comparisons between the performances of parallel flow fields and interdigitated flow fields during cold start-up were made [23,24] Duy et al. simulated the different effects between a sub-channel, a bypass serpentine flow field, and a conventional design on the PEM fuel cell performance by using a combination of different flow field forms at the anode and cathode sides, with results indicating that the sub-channel and the bypass serpentine flow field showed better performances than the conventional design [25].

Compared with a traditional PEM fuel cell (operating temperature less than 100 °C), the HT-PEM fuel cell (operating temperature is around 150–200 °C) is considered to be the next-generation fuel cell, as it has the advantages of fast reaction kinetics, a high tolerance to CO poisoning, and simple water and heat management [26–30] Based on the above research, this work proposes a new serpentine flow field design with rib grooves (namely grooving on ribs) in the PEM fuel cell at 160 °C, and the magnitude order of the grooves' size is between the magnitude order of the flow field and the gas diffusion layer. The distribution of rib grooves can be adjusted according to the design. This kind of flow field structure has not yet been described in simulation or experimental studies.

## 2. Model Development

### 2.1. Model Geometry

Multiphysical field direct coupling analysis software COMSOL Multiphysics was selected to build and compute the PEM fuel cell models with different rib grooves. The schematic diagram of

the modeling domains of the serpentine flow field PEM fuel cell is shown in Figure 1. The geometry was built with seven different layers from bottom to top, which are the anode flow field (A-FF), the anode gas diffusion layer (A-GDL), the anode catalyst layer (A-CL), the membrane (MEM), the cathode catalyst layer (C-CL), the cathode gas diffusion layer(C-GDL), and the cathode flow field (C-FF), respectively.

As shown in Figure 2, in this work, the simulations were conducted using four types of serpentine flow field design with different rib grooves: (a) the conventional design, where the rib groove rate was 0, (b) 1/3 of the rib groove, where the rib groove rate was 1/3, (c) 2/3 of the rib groove, where the rib groove rate was 2/3, (d) full rib groove, where the rib groove rate was 1. To simplify the models, the rib grooves were only added to the cathode flow field. The rib grooves were established along the upper surface of the cathode gas diffusion layer. The depth of the rib groove was 0.1 mm, the angle of rib groove direction with the inlet direction was 45°, and the distance between adjacent rib grooves was 2 mm. The geometric parameters of the PEM fuel cell model used in the simulation are listed in Table 1.

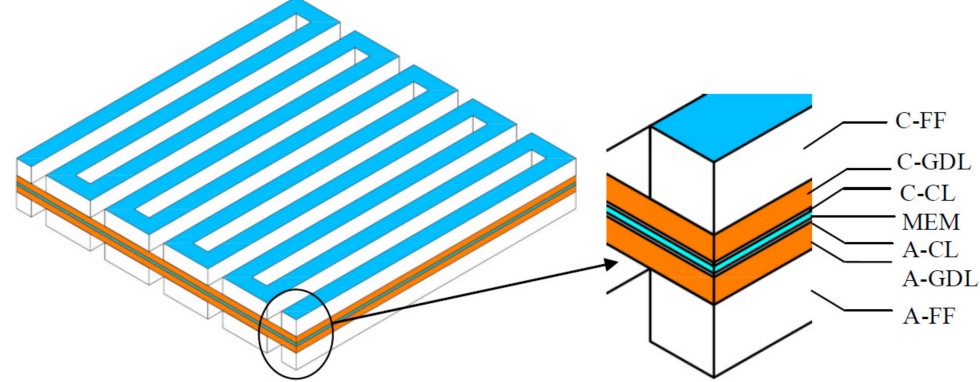

**Figure 1.** Geometry of the fuel cell with a serpentine flow field.

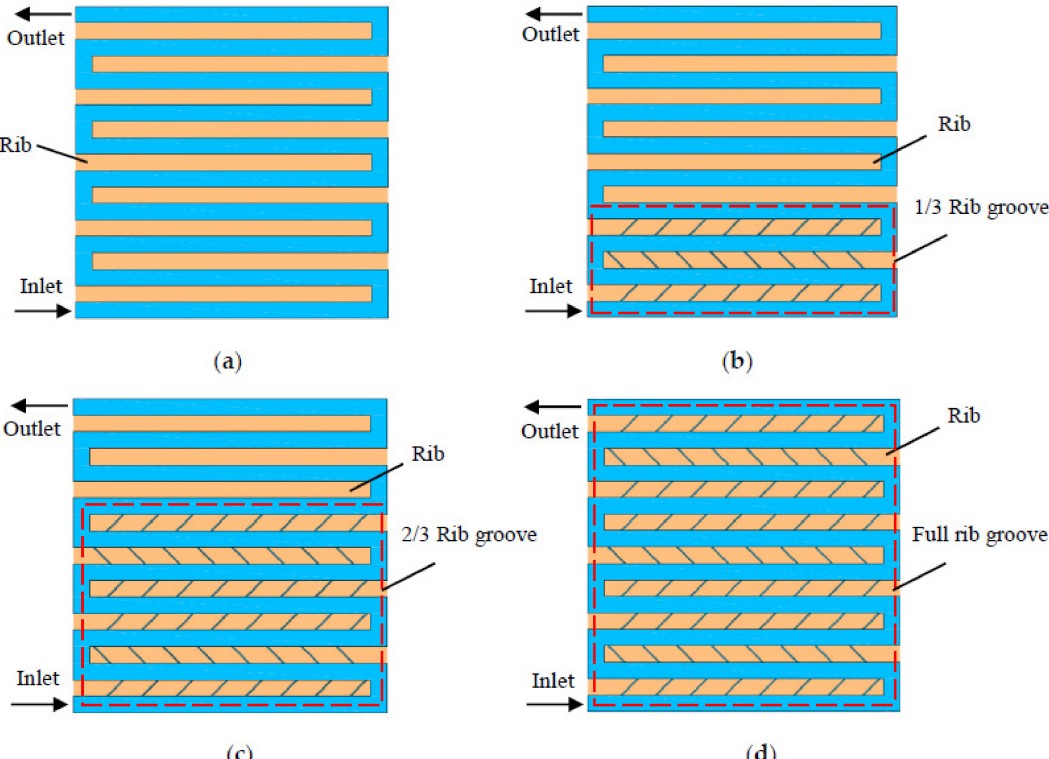

**Figure 2.** Geometry of the serpentine flow field. (**a**) Conventional design, (**b**) 1/3 of the rib groove, (**c**) 2/3 of the rib groove, (**d**) full rib groove.

**Table 1.** Geometric parameters used in the simulation.

| Parameter | Value | Units |
|---|---|---|
| Active area | 3.61 | $cm^2$ |
| Length of the flow field | 19 | mm |
| Width of the flow field | 1 | mm |
| Depth of the flow field | 1 | mm |
| Width of the rib | 1 | mm |
| Thickness of the GDL | 0.38 | mm |
| Thickness of the CL | 0.05 | mm |
| Thickness of the membrane | 0.1 | mm |
| Width and depth of the rib grooves | 0.1 | mm |

*2.2. Governing Equations*

The models were calculated using the Batteries and Fuel Cells Module of COMSOL Multiphysics, and physical fields, such as Secondary Current Distribution, Brinkman Equations, and Transportation of Concentrated Species, were used for the computation. The following are four assumptions that were made to reduce the complexity of the models, which are usually taken as negligible factors for a PEM fuel cell model [11,30]:

(1) The porous media was isotropic and continuous;
(2) The flow was assumed to be laminar inside the fuel cell due to the small Reynolds number in the flow field;
(3) The fuel cells were insulated from the environment and worked at steady state conditions;
(4) Reactant gases were ideal, incompressible, and had non-turbulent flow.

The Navier–Stokes equations were applied to determine the fluid transport in the porous medium and channel.

The continuity equation (mass conservation) and momentum equation (momentum conservation) in the channel were written as

$$\nabla \cdot (\rho u) = 0,  \tag{1}$$

$$-\nabla P + \nabla\left[\mu\left(\nabla u + (\nabla u)^T\right) - \frac{2}{3}\mu(\nabla u)\right] + F_{ex} = \rho(u * \nabla)u,  \tag{2}$$

where $\rho$ is the fluid density, $u$ is the fluid velocity in the channel, $\nabla$ is the Laplace operator, $P$ is pressure, $\mu$ is the dynamic viscosity, and $F_{ex}$ is the force term to include the influence of gravity and other volume forces.

In the porous medium (Brinkman–Darcy's law),

$$\nabla \cdot (\rho v) = 0,  \tag{3}$$

$$-\nabla P + \nabla\left[\frac{\mu}{\varepsilon}\left(\nabla v + (\nabla v)^T\right) - \frac{2}{3}\mu(\nabla v)\right] - \frac{\mu}{k}v + F_{ex} = \frac{\rho}{\varepsilon}(v * \nabla)\frac{v}{\varepsilon},  \tag{4}$$

where $\varepsilon$ is the porosity of gas diffusion layer (GDL), $v$ is the fluid velocity in the porous medium, and $k$ is the effective permeability of GDL.

The Tafel equation was used to describe the reaction kinetics at the cathode side:

$$j = j_{0,c}\left(-\frac{C_{O_2}}{C_{O_2}^{ref}} * 10^{\frac{\eta}{A_c}}\right),  \tag{5}$$

where $j$ is the exchange current density, $j_{0,c}$ is the cathode reference exchange current density, $C_{O_2}$ is the oxygen concentration, $C_{O_2}^{ref}$ is the oxygen reference concentration, $A_c$ is the cathodic slope, and $\eta$ is the cathode over-potential, which was shown as:

$$\eta = \phi_s - \phi_m - E_{ocv}, \tag{6}$$

where $E_{ocv}$ is the open circuit voltage, $\phi_s$ is the solid phase potential, and $\phi_m$ is the membrane phase potential.

The linearized Bulter–Volmer Equation was used to describe the reaction kinetics at the cathode side:

$$j = j_{0,a} \left( -\frac{C_{H_2}}{C_{H_2}^{ref}} \right)^{0.5} \left( \frac{(\alpha_a + \alpha_c)F}{RT} \right) \eta, \tag{7}$$

where $j$ is the exchange current density, $j_{0,a}$ is the anode reference exchange current density, $C_{H_2}$ is the hydrogen concentration, $C_{H_2}^{ref}$ is the hydrogen reference concentration, $\alpha_a$ is the anode transfer coefficient, $\alpha_c$ is the cathode transfer coefficient, $R$ is the gas constant, $T$ is the gas temperature, and $F$ is the Faraday constant. More details about the key physiochemical parameters used in the simulation are shown in Table 2, and the operation temperature was taken as 160 °C.

**Table 2.** Key physiochemical parameters used in the simulation.

| Parameter | Value | Units |
|---|---|---|
| GDL porosity | 0.4 | – |
| GDL electric conductivity | 222 [31] | S/m |
| Inlet $O_2$ mass fraction (cathode) | 0.228 [31] | – |
| Inlet $H_2O$ mass fraction (cathode) | 0.023 [31] | – |
| Anode inlet flow velocity | 0.2 | m/s |
| Cathode inlet flow velocity | 0.5 | m/s |
| Anode viscosity | $1.19 \times 10^{-5}$ [31] | Pa*s |
| Cathode viscosity | $2.46 \times 10^{-5}$ [31] | Pa*s |
| Reference pressure | $1.01 \times 10^5$ | Pa |
| Cell temperature | 433.15 | K |
| Hydrogen reference concentration | 40.88 [31] | mol/m$^3$ |
| Oxygen reference concentration | 40.88 [31] | mol/m$^3$ |
| Membrane conductivity | 9.825 [32] | S/m |

*2.3. Model Verification*

In order to verify the reliability of the PEM fuel cell model, the simulated data was compared with the experimental data under the same operating conditions using a single flow field PEM fuel cell, of which the length was 40 mm. The experimental conditions were as follows: the operation temperature was 160 °C, with no pre-humidification for hydrogen and air, the flow rate of hydrogen and air were 100 mL/min and 200 mL/min, respectively, and the back pressure was 101 kPa [30,33]. As can be seen in Figure 3, under the high and medium cell voltage conditions, the polarization and power density curves of the simulated results matched very well with the experimental results; when the cell voltage was low, the experimental results were slightly higher than the simulated results. However, on the whole, there was good agreement between the simulation data and the experimental results. Hence, the reliability of the mathematical model was proven. In addition, grid-independency tests were conducted and the results were independent.

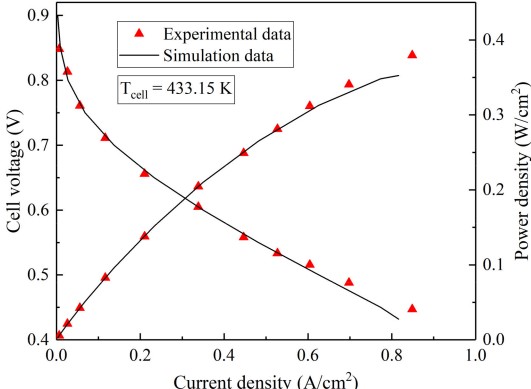

**Figure 3.** Comparison between the simulation data and the experimental data.

## 3. Results and Discussion

### 3.1. Effect of Temperature

Figure 4 illustrates the polarization and power density curves of the PEM fuel cell models with serpentine flow fields at various operating temperatures ranging from 393.15 to 453.15 K. As seen from Figure 4, the output performance of the PEM fuel cell models improved as the temperature increased. When the temperature was higher than 433.15 K, the improvement of the output performance became lower, as the performance curve at 433.15 K was very close to that at 453.15 K. Therefore, in this study, 433.15 K was selected as the operating temperature for different PEM fuel cell models.

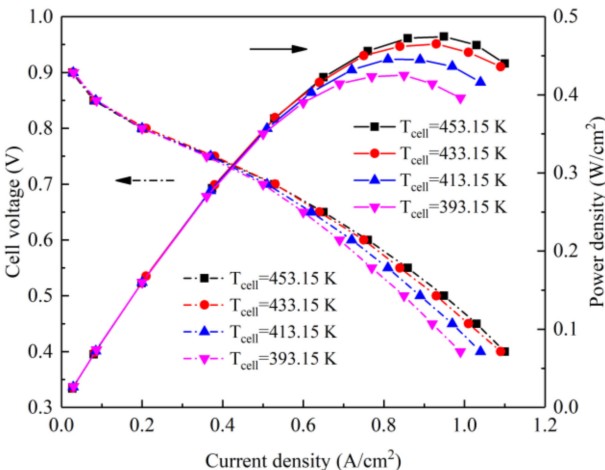

**Figure 4.** Polarization and power density curves at different temperatures.

### 3.2. Effect of Rib Groove Designs

Figure 5 illustrates the polarization and power density curves of the conventional design and the different rib groove designs. Figure 5 shows that, when the operating voltage of the fuel cell was less than 0.8 V, as the voltage decreased, the current density and power density under the rib grooving (1/3 of the rib groove, 2/3 of the rib groove, and full rib groove) were greater than the same values under the conventional design. Furthermore, we also found that when the rib groove rate increased from 0 to 1, the performance of the fuel cells continued to improve. However, when the rib groove rate was greater than 2/3, there was less improvement of the fuel cell performance. When the operating voltage was lower than 0.6 V, the performance of the 2/3 rib groove was almost the same as that of the full rib groove. This was because as the operating voltage decreased and the current density increased, more oxygen on the cathode side was consumed, so the diffusion of oxygen in the cathode gas diffusion layer and catalyst layer became a key factor affecting the output performance of the fuel cell. The grooving

ribs improved the flow consistency of the reactant and product behavior in the flow fields, and then enhanced the current density in the ohmic resistance loss and mass transport loss regions [25]. Figure 6 shows the oxygen concentration at the interface of the cathode GDL and catalyst layer (CL) when the voltage was 0.6 V. Figure 6 shows that, in the case of rib grooving designs, the concentrations of oxygen in the GDL and CL on the cathode side were significantly greater than in the conventional design, thereby showing an improvement in both the utilization rate of catalyst and the output performance of the fuel cell. While the oxygen concentration was almost the same when the rib groove rate was 2/3 and 1, the phenomenon of the similar output performance between the two cases was further explained.

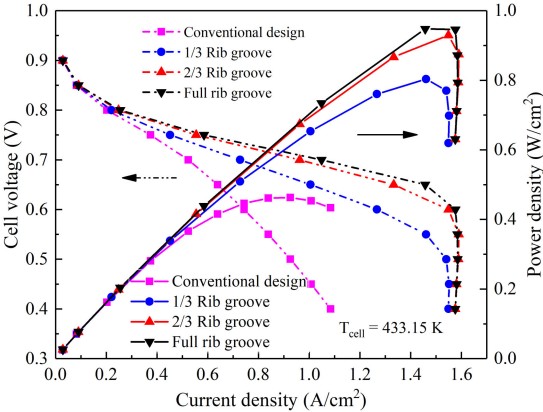

**Figure 5.** Polarization and power density curves of conventional and different rib groove designs.

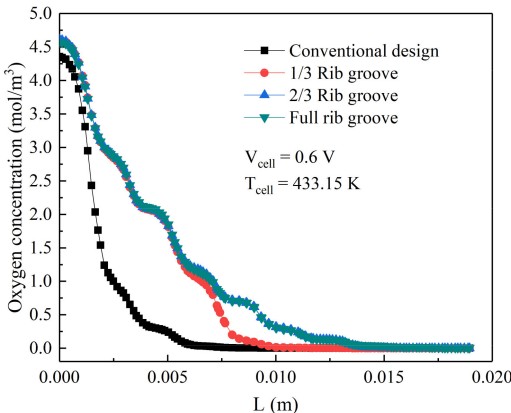

**Figure 6.** Oxygen concentration at the interface of the cathode gas diffusion layer (GDL) and catalyst layer (CL).

Figure 7 shows the membrane current density of the middle surface of the proton exchange membrane under the conventional design and different rib groove designs when the cell voltage was 0.6 V. As can be seen in Figure 7, in all four cases, the current density decreased gradually along the gas flow direction, because the reactants were consumed gradually. However, in general, the membrane current density of the rib grooving flow fields of the PEM fuel cells was more uniform than in the conventional design. With the increase of the rib groove rate from 0 to 1, the minimum membrane current densities were 20, 30, $1.72 \times 10^3$, and $3.66 \times 10^4$ A/m$^2$ for the conventional design, 1/3 of the rib groove, 2/3 of the rib groove, and the full rib groove, respectively. As a result, the overall performance of the PEM fuel cell was improved when the rib grooving designs were applied.

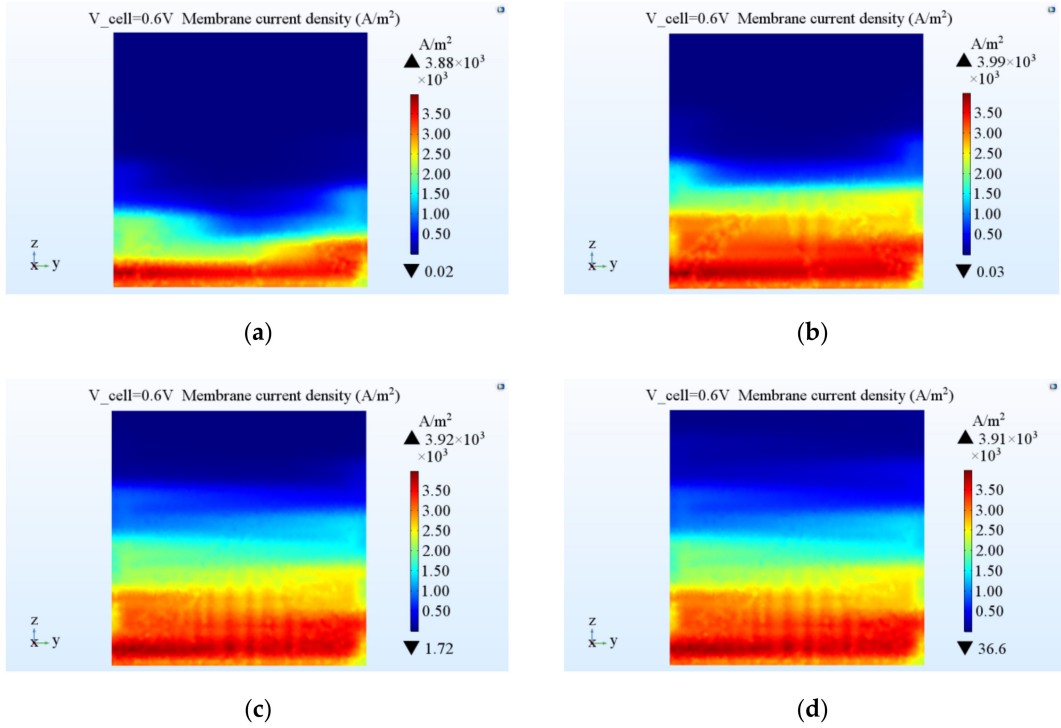

**Figure 7.** Membrane current density when the cell voltage was 0.6 V. (**a**) Conventional design, (**b**) 1/3 of the rib groove, (**c**) 2/3 of the rib groove, (**d**) full rib groove.

### 3.3. Distribution of Oxygen Concentration at the Cathode Side

Figures 8 and 9 show the oxygen concentration distribution at the interface between the cathode flow field and the gas diffusion layer and the oxygen concentration distribution from the inlet to the outlet along the cathode flow field, respectively, when the cell voltage was 0.6 V. As can be seen in Figures 8 and 9, due to consumption of the reactants by the electrochemical reaction, the oxygen concentration gradually decreased along the direction of the gas flow. This caused a low concentration of oxygen near the outlet, then leading to local reactant starvation and low fuel cell performance. It can be seen in Figure 8 that, with the increase in the rib groove rate from 0 to 1, the oxygen distribution in the serpentine flow fields became more and more uniform due to the gas pressure at the entrance being high, thereby allowing the rib groove structure on the flow field to alleviate the uneven distribution of oxygen in the flow field. Figure 9 shows that the oxygen concentration in the convention serpentine design flow field first drastically reduced and dropped slowly after the third corner, but for serpentine flow fields with rib grooves, the oxygen concentration in the flow field had no obvious corner characteristics, instead decreasing slowly and showing greater stability. The oxygen concentration distribution was similar when the rib groove rate was 2/3 and 1, both of which were better than that of 1/3 of the rib groove rate. The rib groove structure reduced the concentration loss caused by insufficient oxygen reactants in the catalyst layer and provided a more uniform current density distribution, thus improving the overall performance of the PEM fuel cell.

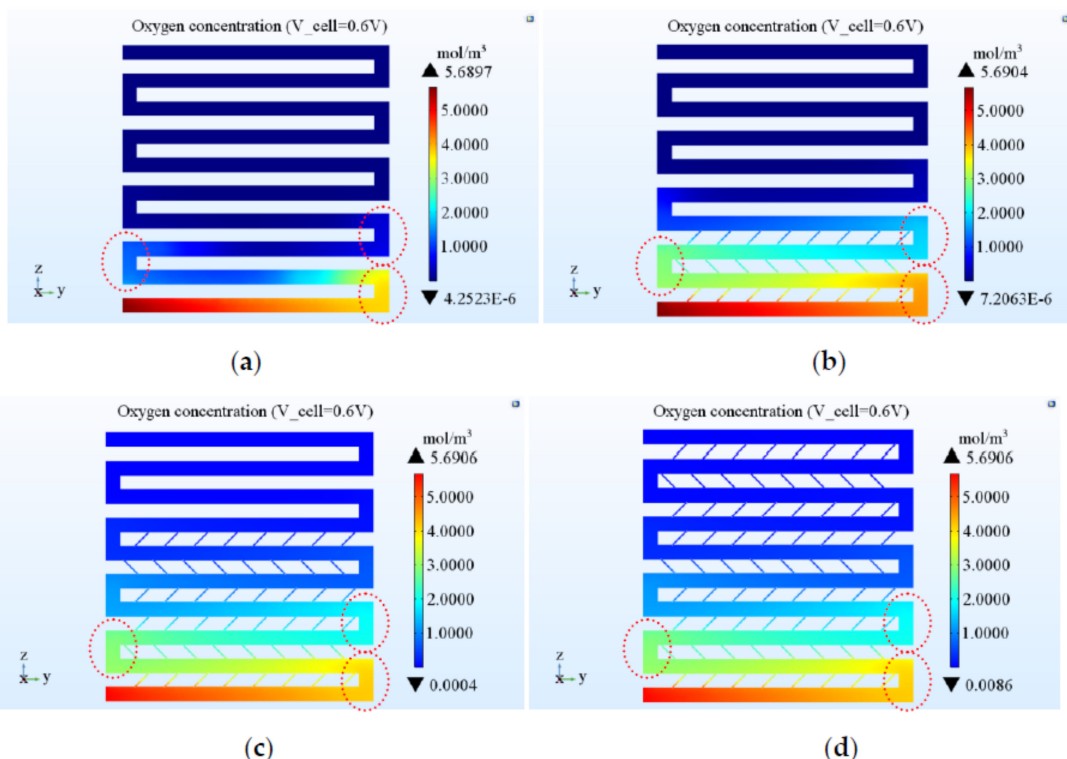

**Figure 8.** Oxygen concentration of the cathode flow field when the cell voltage was 0.6 V. (**a**) Conventional design, (**b**) 1/3 of the rib groove, (**c**) 2/3 of the rib groove, (**d**) full rib groove.

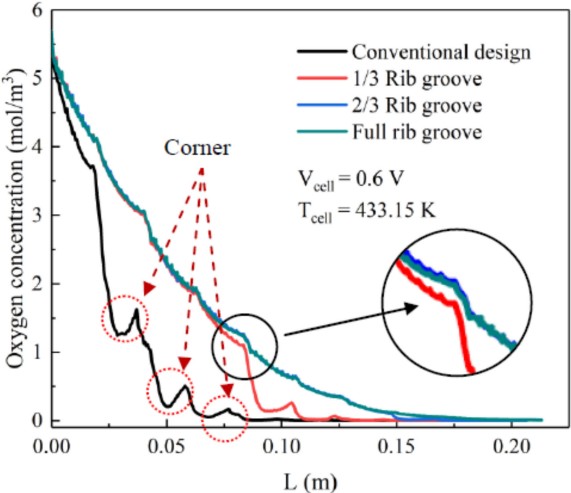

**Figure 9.** Oxygen concentration along the cathode flow field.

### 3.4. Distribution of Water Vapor

Figure 10 shows the concentration of water vapor in the gas diffusion layer under the rib on the cathode side, for which similar studies have been carried out previously [22,30,31]. It can be clearly seen from Figure 10 that the concentration of water vapor under the rib of the serpentine flow field in the conventional design was significantly higher than that in the rib groove designs, and the water vapor concentration under the rib was higher in the middle and lower on both sides. This was because the water vapour generated by the electrochemical reaction in the conventional design of serpentine flow field PEM fuel cell could only be discharged along the direction of the flow field, where the flow path was long and the resistance was large. However, the water vapor under the rib of the serpentine flow field with a rib groove design could also be transported through the grooves, reducing drainage

resistance and therefore enhancing the drainage capacity of the fuel cell, thereby simplifying the water management system of the PEM fuel cell.

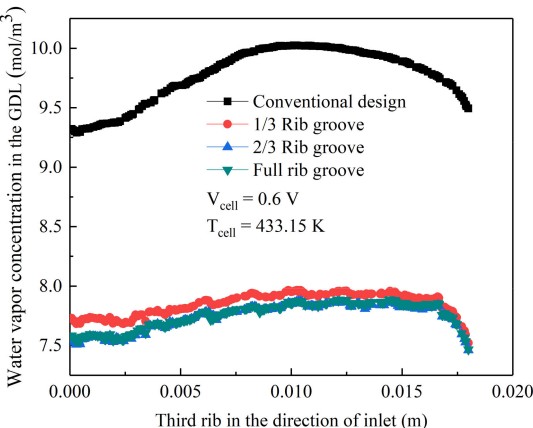

**Figure 10.** Water vapor concentration under the rib.

## 4. Conclusions

In order to improve the output performance of PEM fuel cell with the serpentine flow field, a new rib grooving serpentine flow field design was proposed. The numerical results demonstrated that a higher current and power density were obtained by increasing the rib groove rate of the serpentine flow field. Compared with the conventional design, the new design enhanced the distribution uniformity of the oxygen concentration in the flow field, reduced the water vapor concentration under the rib, and increased the oxygen concentration at the interface between the gas diffusion layer and the catalyst layer in the cathode. When the rib groove rate increased from 0 to 1, the distribution uniformity of the oxygen along the cathode flow field, the oxygen concentration at the interface between the gas diffusion layer and the catalyst layer, and the current density of the proton exchange membrane were all improved. When the rib groove rate was 1, the maximum current density and the highest power output value were obtained.

**Author Contributions:** Conceptualization, S.C. and W.Y.; data curation, Z.X.; formal analysis, X.L.; funding acquisition, S.C.; investigation, Z.X.; methodology, S.C.; resources, X.Z.; software, X.L., Z.X. and X.Z.; supervision, S.C.; writing—original draft, X.L., Z.X. and W.Y.; writing—review and editing, X.Z. and Y.W.

**Funding:** This research was funded by Natural Science Foundation of Liaoning Province, NO: 20170540734.

**Acknowledgments:** The authors would like to thank the Analysis and Testing Center of Shenyang Jianzhu University for offering us their computing facilities.

**Conflicts of Interest:** The authors declare no conflict of interest.

## Nomenclature

| | | | |
|---|---|---|---|
| $\rho$ | fluid density (kg/m$^3$) | $\mu$ | dynamic viscosity (Pa*s) |
| $v$ | fluid velocity in the porous medium (m/s) | $\varepsilon$ | porosity of GDL |
| $\nabla$ | Laplace operator | $k$ | effective permeability of GDL (m$^2$) |
| $P$ | pressure (Pa) | $u$ | fluid velocity in the flow field (m/s) |
| $j$ | exchange current density (A/m$^2$) | $\alpha_c$ | cathode transfer coefficient |
| $j_0$ | reference exchange current density (A/m$^2$) | $\eta$ | activation over-potential (V) |
| $n$ | number of electrons transfer in the reaction | $\delta$ | GDL thickness (m) |
| $T$ | absolute temperature (K) | $R$ | universal gas constant |
| $C_{O_2}^{ref}$ | reference oxygen concentration (mol/m$^3$) | $C_{O_2}$ | oxygen concentration (mol/m$^3$) |
| $E_{ocv}$ | open circuit voltage (V) | $F$ | Faraday constant |
| $E_{cell}$ | operating voltage of the fuel cell (V) | $j_L$ | limiting current density (A/m$^2$) |
| $R_{ohm}$ | lumped resistance ($\Omega$*m$^2$) | $D^{eff}$ | effective diffusion coefficient |

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
