# Peer review of "Numerical Simulation of a New Flow Field Design with Rib Grooves for a Proton Exchange Membrane Fuel Cell with a Serpentine Flow Field"

_applsci, doi:10.3390/app9224863_

Round 1
Reviewer 1 Report
For high temperature (120-200oC) PEMFC, such as PBI system, no water is needed both in cathode inlet O2 (air) and anode inlet H2 gases. Also the water vapor product in the cathode is evaporated under high temperature (160C in the present manuscript), no water presents in the cathode GDL and flow field. The authors should study carefully the related high temperature PEMFC papers shown in the references cited in the manuscript.
Reviewer 2 Report
The authors investigated new flow design with rib grooves for PEMFC. The simulation results are interesting and very helpful for those who working on improving proton exchange membrane fuel cell performance. I have few comments that authors should consider or address in the revised manuscript.
Though, authors studied and performed simulation calculation at 160 C for high temperature next generation fuel cell applications it would be interesting to see the results of new flow design effects on fuel cell performance at various temperature. So author should include new temperature dependent fuel cell performance results with new flow design in the revised manuscript. Typo, Page 2, line 44, "4-5678"
Round 2
Reviewer 1 Report
The manuscript can be published in the present form.
This manuscript is a resubmission of an earlier submission. The following is a list of the peer review reports and author responses from that submission.
Round 1
Reviewer 1 Report
[1] Fig. 1--- i-V and i-power density experimental data; What were the experimental conditions? Such as temperature, humidity, O2 flow rate, H2 flow rate, back pressure et.?
[2] Fig. 9--- Normally the working temperature for high temperature proton exchange membrane fuel cells (HT-PEMFCs) is around 150-200C. Under the high temperature (Temp> 100C), usually the cathode product, i.e., water, is evaporated and does not cumulate in the gas flow field. Comparing to the Nafion-based low temperature PEMFCs (working temperature < 100C), the very low water flooding in cathode flow channel is the main advantage of HT-PEMFC. How did the authors detect "water" under the rib. What was the simulated fuel cell working temperature.
[3] Table 2--- Inlet H2O mass fraction - H2O was in the cathode or in the anode flow field? When working at high temperature (Temp > 100C), most of the H2O is evaporated.
Reviewer 2 Report
The title of the manuscript says "...high temperature proton exchange membrane...."; however, this study shows no relation to high temperature proton exchange membrane fuel cell. Some phenomena of the HT-PEMFC were not captured in this study.2. Reference [30] was used for model validation; however, reference [30] conducted experiments using a low temperature PEMFC.
3. In Fig. 4, the current density at 0.6 V, showing higher than 1 A/cm2, is better than a low temperature fuel cell. This is not reasonable.